Occurrence of microplastics in edible aquatic insect Pantala sp. (Odonata: Libellulidae) from rice fields

Maneechan Witwisitpong 1
Prommi Taeng On faastop@ku.ac.th 2
1 Program of Bioproducts Science, Department of Science, Faculty of Liberal Arts and Science, Kasetsart University, Kamphaeng Saen Campus , Nakhon Pathom , Thailand
2 Department of Science, Faculty of Liberal Arts and Science, Kasetsart University, Kamphaeng Saen Campus , Nakhon Pathom , Thailand
Oehlmann Jörg
Electronic publication date: 2022 Feb 8
Publication date: 2022
Volume: 10
Electronic Location ID: e12902
Received 2021 Oct 13; Accepted 2022 Jan 17
Copyright: ©2022 Maneechan and Prommi
Copyright year: 2022
Copyright holder: Maneechan and Prommi
License: This is an open access article distributed under the terms of the Creative Commons Attribution License, which permits unrestricted use, distribution, reproduction and adaptation in any medium and for any purpose provided that it is properly attributed. For attribution, the original author(s), title, publication source (PeerJ) and either DOI or URL of the article must be cited.
License URL: https://creativecommons.org/licenses/by/4.0/

Keywords: Edible aquatic insects, Odonata, Rice field, Microplastics, FTIR

Funding: National Research Council of Thailand (NRCT) NRCT5-RGJ63002-041 This research project is supported by the National Research Council of Thailand (NRCT): NRCT5-RGJ63002-041. The funders had no role in study design, data collection and analysis, decision to publish, or preparation of the manuscript.

==============================
Background

Microplastic (MP) contamination has been discovered in aquatic systems throughout the world. They are well known as contaminants in aquatic species, but there is a gap in understanding about pathways of MP contamination into humans (i.e., through aquatic animals). The goal of this study is to assess MP contamination in an edible aquatic insect (Pantala sp.) living in rice fields.

Methods

A dragonfly larva, Pantala sp. (Odonata: Libellulidae), was tested for MPs. The study concentrated on three distinct anatomical compartments (whole body, gastrointestinal tract, and body without gastrointestinal tract), each of which was examined separately. For the physical identification and chemical analysis of MPs, a stereomicroscope and a Fourier transformed infrared spectroscope (FT-IR) were used, respectively.

Results and Discussion

The microplastics content was 121 in the whole body, 95 in the gastrointestinal tract, and 66 in the body without the gastrointestinal tract, with an average of 1.34 ± 1.11, 1.06 ± 0.77, and 0.73 ± 0.51 abundance/ individual, respectively. The most common MPs discovered during this study were fragments, followed by fibers and rods. The chemical analysis by FT-IR confirmed three different polymers, including polymethyl methacrylate (PMMA), polyethylene terephthalate (PET), and polypropylene (PP). There was no significant difference in MP abundances among the sample types (Kruskal-Wallis chi-squared = 2.774, df = 2, p = 0.250). The findings suggest that eating an edible aquatic insect (Odonata: Pantala sp.) could be one way for humans to ingest MPs.

Introduction

Plastic use and misuse has been noted as a major environmental issue in both aquatic and terrestrial environments (Wright, Thompson & Galloway, 2013; Bläsing & Amelung, 2018). The continuous increase in the discharge of plastics into the environment, as a result of the growing human population, is one of the main sources of plastic pollution (Thompson et al., 2004; Nel et al., 2017). Microplastics (MPs) are plastic particles with a diameter of less than 5 mm (GESAMP, 2016). These MPs are generally categorized as primary MPs (manufactured for addition to certain products) or secondary MPs (derived from the breakdown of larger plastics) based on their structure and chemical composition (Cole et al., 2011; Ehlers, Manz & Koop, 2019; Arthur, Baker & Bamford, 2009). Microplastic pollution has been reported in a variety of environments and species across the world. Although numerous studies have been conducted to explore the occurrence, abundance, and dispersion of MPs in the marine environment, only a few have focused on microplastics in freshwater habitats (Free et al., 2014; Horton et al., 2017; Mani et al., 2015; Castañeda et al., 2014; Dris et al., 2015). Microplastic debris contaminates freshwater habitats such as streams, rivers, and lakes (Eerkes-Medrano, Thompson & Aldridge, 2015). Plastic contamination can have an impact on the organisms being exposed. Microplastics can be consumed by a wide range of animals from zooplankton to large vertebrates, and they are primarily accumulated in the stomach (Qiao et al., 2019). Therefore, concerns of MP contamination are growing rapidly since ingestion is more likely in lower trophic organisms, which transfers via the food chain (Al-Jaibachi, 2019).

Rice field ecosystems function as temporary wetlands, which are a unique man-made environment that connects and shares water with natural wetlands (Al-Shami et al., 2010; Lutz, Kehr & Fernández, 2015; Wakhid et al., 2020). According to Heckman (1974), who recorded a total of 589 species of organisms on a rice field over the period of a year, including edible aquatic insects, rice fields contain a remarkable high biodiversity. Insects with aquatic larvae have been documented as human food in 48 countries throughout the world. Macadam & Stockan (2017) reported that the Coleoptera have the most edible food insects (79 species) utilized, followed by Odonata (58 species) and Hemiptera (55 species).

An overview of the nutritional makeup of Odonata insects was shown in a previous study (Feng et al., 2001; Xiaoming et al., 2010; Narzari, Sarmah & Gupta, 2017). The nymphal stage of aquatic insects such as dragonflies (Order Odonata) can be consumed because it is easier to capture than the adult form. The larvae contain all of the nutrients, including protein, lipids, amino acids, and microelements. Dragonfly nymphs (Libellulidae, Aeshnidae, Gomphidae) are commonly consumed in China (Ying et al., 2001; Macadam & Stockan, 2017; Williams & Williams, 2017), India (Chakravorty, Ghosh & Meyer-Rochow, 2013), the Philippines (DeFoliart, 1992), Laos (Pemberton, 1995; Hanboonsong & Durst, 2014; Barennes, Phimmasane & Rajaonarivo, 2015), and Thailand (Hanboonsong, 2010).

It is well known that edible aquatic insects with high nutritional content are consumed in Thailand. The preference for Libellulidae (Pantala sp.) species in the Northern and Northeastern parts of this country is remarkable. These nymphs are pale greenish with light brown markings (Bright, 2010). They are aggressive, fast-growing predators that adapt well to lentic habitats, including man-made habitats. Although this species has a worldwide distribution, it is uncommon in Europe (Kiany & Minaei, 2010; Günther, 2019). Based on the habitat, a dragonfly larva (Pantala sp.) may ingest MPs for several reasons. They feed on a wide variety of prey, including zooplankton and smaller macroinvertebrates (Byers, 1940; Lamb, 1924; Warren, 1915). According to the findings on MPs in freshwater sediment reviewed by Yang et al. (2021), dragonfly larvae may confuse MP particles with prey and ingest them. Thus, these MP pollutants can be transferred to Pantala sp. and other predators such as fish, birds, and humans. This is concerning as we would expect there to be double the number of MPs in the whole body as it includes both the gastrointestinal tract and other tissues. The goal of this study is to investigate MP contamination in an edible aquatic dragonfly, Pantala sp. (Odonata: Libellulidae), that is consumed by people in many parts of Thailand.

Materials & Methods

Study area

Samples were taken from a rice field in the Kasetsart University Kamphaeng Saen campus, Kamphaeng Saen district, Nakhon Pathom province, central Thailand (N14°00′32.2474 E99°58′54.1744) (Fig. 1).

Figure 1 Site of sampling collection (N14°00′32.2474 E99°58′54.1744).

Collection and identification of samples

Aquatic insects were sampled at random along the edge of the rice plots in October 2020 using an aquatic dip net (dimensions 30 × 30 cm, 250 m mesh, 50 cm length). According to our preliminary samplings, several aquatic insects were discovered towards the margin of the rice plot, due to the greater volume of water. As a result, this section of the rice plot was sampled. For such a rice plot, samples were taken along 3 m on each side. Aquatic insect collections were made in each plot by dragging a dip net down the ground for 3 m along the margin of the rice plot. Because aquatic insects have high water content, aquatic insects caught in the net were collected and preserved in vials containing 95% alcohol. The standard keys were used to identify aquatic insects up to the lowest taxonomic level (Dudgeon, 1999; Yule & Yong, 2004). The nymph of Pantala sp. (Odonata, Libellulidae) was used in this research as an aquatic insect for MP analysis (Fig. 2). In addition, more Pantala sp. were collected in the same rice field for the MPs’ investigation in October 2021.

Figure 2 (A–B) Morphology of Pantala sp. (Libellulidae) nymph; (C–D) deep fried chicken egg with nymph, a popular northern Thai meal.

Preparation of Pantala sp. larvae

Only the Pantala sp. taxa were used for the microplastic investigation. For the study, a total of 180 Pantala sp. specimens of comparable weight were analyzed. Three times distilled water was used to wash the fresh preserved specimens. To assess MPs from Pantala sp., 5 specimens (×18 replicates) of the whole body, 5 specimens (×18 replicates) of the gastrointestinal tract (GT) alone, and 5 specimens (×18 replicates) of the body without the GT were pooled. The specimens were dissected individually using scissors and forceps. All replicates were transferred to a 25 mL erlenmeyer flask. The entire body, the GT, and the body without the GT weight were all measured for wet weight. The results were presented as a mean ± standard deviation.

H2O2 treatment

Each pooled sample was placed in one erlenmeyer flask and eighteen replicates were prepared for each sample type. To digest the organic materials, 10 mL of a hydrogen peroxide solution (30% H2O2) was added to each flask (Ehlers, Manz & Koop, 2019). After that, the flasks were wrapped in parafilm and stored at room temperature for 7 days.

Floatation and filtration

Following tissue disintegration, potassium formate (99%) was employed for density of separation the resulting dissolved liquid from soft tissues (Ehlers, Manz & Koop, 2019). Each sample was placed in a glass separatory funnel, which was then filled with approximately 16 g of potassium formate and shaken to separate the solution. Because of its less dense form, saturated potassium formate solution facilitates the separation of the microplastic layer after about 4 h. The undissolved inorganic residues were then drained, and the supernatant was vacuum filtered onto nylon membrane filters (pore size of 0.45 µm; diameter of 47 mm). The filter was then placed in clean glass petri dishes with aluminum foil covers and dried for two days in a 50 °C drying cabinet.

Contamination control

To avoid contamination from airborne MPs, all containers and equipment were cleaned with distilled water and covered with aluminum foil when not in use. Exclusive gloves (nitrile), steel, and glass devices were always used at the laboratory. Before beginning labwork, the scissors and forceps were rinsed three times with deionized water. Lab surfaces were thoroughly cleaned with 70% ethanol. At every stage of the analysis, blanks were run without tissues in parallel with the same procedure used for the samples. All the experimental procedures were finished as soon as possible.

Microplastic observation and polymer identification

Under a stereomicroscope (Leica EZ4E) with 35x magnification, the filters were visually examined, and photos were obtained at various magnifications to identify MP particle based on their color and type. The MP particles were recorded. A PerkinElmer Spectrum-Fourier transform infrared spectrometer (FT-IR) in attenuated total reflection (ATR) mode was used to verify selected particles (range size 400–500 µm). The spectral range was 4,000 to 500 cm−1, with a 32 cm−1 spectral resolution and 16 co-scans for each measurement. The characterization of functional groups and the analysis of polymer types were compared to the Bruker spectrum library. Considering the spectrum analysis, the matching degree of spectra with a quality index ≥0.7 was accepted (Woodall et al., 2014).

Data analysis

The abundance, types, and colors of MPs were counted. Non-parametric analyses were applied as the abundance of MPs was not a normal distribution among the three sample types. Kruskal–Wallis H test was used to test for differences in the mean abundance of MP between sample types, using the Statistical Package for the Social Sciences (SPSS) version 19. Statistical significance was defined as a p-value of less than 0.05 (p < 0.05).

Results and Discussion

Abundance, type and color of MP in edible aquatic insects

The mean wet weights of Pantala sp. whole body, gastrointestinal (GT) tract only, and body without the GT were 0.3098 ± 0.0795, 0.0399 ± 0.0133 and 0.2445 ± 0.0707 g, respectively. In the controls, no MP particles were found in the blanks. The total number of particles were 121, 95, and 66 in all eighteen replicates of pooled samples (Figs. 3A–3C), with mean abundance per individual of 1.34 ± 1.11, 1.06 ± 0.77 and 0.73 ± 0.51 items in the whole body, gastrointestinal (GT) tract, and body without the GT, respectively. Kruskal-Wallis H test revealed no significant differences in the mean abundance of MP particles among sample types (chi-squared = 2.774, df = 2, p = 0.250) (Table 1). Different types of MPs identified in three samples were fragment, fiber, and rod (Figs. 3D–3F). The colors of MPs were shown in five different shades of red, green, blue, violet, and orange (Figs. 3G–3I).

Figure 3 Comparison of the abundance (A–C), type (D–F), and color (G–I) of MPs in Pantala sp. abbreviation: C, control.

Table 1 MPs inspected in the three sample types.

Sample type	Wet weight (g)	Total number of particles	Mean abundance /individual*	
Whole body (n = 90)	0.3098 ± 0.0795	121	1.34 ± 1.11	
Gastrointestinal tract (GT) only (n = 90)	0.0399 ± 0.0133	95	1.06 ± 0.77	
Body without GT (n = 90)	0.2445 ± 0.0707	66	0.73 ± 0.51	
Notes.

* No significant difference at p = 0.250 (Kruskal–Wallis H test).

Fragments and fibers were the most common particle types in edible aquatic insects (Odonata), which was similar to prior observations in Nigerian freshwater insects (Akindele, Ehlers & Koop, 2020). They were found in different sample types in this study, indicating that edible aquatic insects may be vulnerable to MP pollution. The whole bodies had more microplastic items than the other two samples because that is the structure where food and other ingested materials are deposited. In the case of contamination in the body without GT, microplastics could be retained in the exoskeleton of the insect body. Ingestion of MPs, on the other hand, is likely to have different effects on an organism depending on its size, shape, concentrations, and exposure time (Redondo-Hasselerharm et al., 2018).

Identification of microplastic polymers by FT-IR

A selected 52 plastic-like particles (about 18% of total MPs) were identified by a Fourier Transformed Infrared Spectroscope (FT-IR). Some particles were confirmed as polymethyl methacrylate (PMMA), polyethylene terephthalate (PET), and polypropylene (PP) (Table 2). For the 52 selected particles, 46.1% were identified as microplastics, 15.4% as non-microplastics, and 38.5% as unidentified particles. The spectral characteristics of these polymers are shown in Figs. 4A–4C.

Table 2 Types of MPs identified with FTIR.

	Whole body	GT	Body without the GT	Total	
	no.	%	no.	%	no.	%	no.	%	
Particles measured	21	100	14	100	17	100	52	100	
MP particles	10	47.6a	5	35.7	9	52.9	24	46.1	
Polyethylene terephthalate (PET)	7	70.0b	5	100.0	5	55.6	17	70.8	
Polypropylene (PP)	2	20.0	0	0	4	44.4	6	25.0	
Polymethyl methacrylate (PMMA)	1	10.0	0	0	0	0	1	4.2	
Non MP particles	3	14.3	4	28.6	1	5.9	8	15.4	
Cellulose powder	1	33.3	1	25.0	0	0	2	25.0	
Polyethylene glycol	2	66.7	2	50.0	0	0	4	50.0	
Xanthan gum	0	0	1	25.0	0	0	1	12.5	
Hydroxyethyl cellulose	0	0	0	0	1	100.0	1	12.5	
Unidentified particles	8	38.1	5	35.7	7	41.2	20	38.5	
Notes.

a The percentage of MP particles in all the measured particles.

b The percentage of each type in all the MP particles.

Figure 4 FTIR spectra of representative MP polymers.

The red spectrum is that of the FT-IR measurement, while the black spectrum is the reference spectrum from the Bruker spectrum library. The black arrows in the photographs indicate the particles that were identified.

The findings suggest evidence of detecting MPs in aquatic insects such as Pantala sp. (Odonata: Libellulidae) that humans eat from rice fields. Previously, there are limited research on plastic contamination in Thailand, particularly on aquatic insects in freshwater environments. Recent research (Windsor et al., 2019) revealed data on the occurrence of MP particles in aquatic macroinvertebrates in a riverine valley in South Wales, UK, with the presence of MPs in almost half of the samples (0.14 MPs/mg tissue). Ehlers, Manz & Koop (2019) conducted a similar study and discovered that MPs (e.g., polypropylene, polyethylene, and polyvinyl chloride) were present in the biological structure of freshwater organisms. Microplastics can also pass through mosquito life stages (i.e., larva, pupa, and adult) and spread throughout aquatic systems (Rhodes, 2019). Furthermore, ingestion of MP polymers was reported in Chironomus sp. (Diptera) (i.e., styrene ethylene butylene styrene, acrylonitrilebutadiene styrene (ABS), chlorinated polyethylene, polypropylene (PP), and polyester), Siphlonurus sp. (Ephemeroptera) (i.e., polyester and ABS) and Lestes viridis (Odonata) (i.e., polyester and PP) from Ogun and Osun Rivers, Nigeria (Akindele, Ehlers & Koop, 2020). As larger plastic debris breaks into smaller plastic bits, Cole et al. (2011) discovered that MPs were more likely to be fragments. Also, fibers, and rods were generated from original MPs. Secondary MPs make extrapolating results from single-species and virgin MP investigations challenging (Rummel et al., 2016). One of three ways that freshwater systems become contaminated by MPs are via effluent discharge, overflow of wastewater sewers during high rain events, and run-off from sludge, which all occur in rice fields (Eriksen et al., 2013). Storms and extreme weather conditions, according to can aggravate the flow of MPs from land to water (Cole et al., 2011).

In terms of trophic transfer and the potential for effects across multiple trophic levels, D’Souza et al. (2020) showed that plastics can be transported from invertebrate consumers to predators in natural freshwater ecosystems, including humans. Plastic is contaminating practically every area of the world and its ecosystems, which is a startling fact. However, ocean pollution and plastic deposition on marine animals and sea birds have received the most attention so far. As a result, it is past time for us to focus on freshwater sources as well.

Conclusion

This is a pilot study that indicates MPs are found in a larval odonate (Pantala sp.), which is consumed by humans in many countries across the world. Certainly, more research into MP pollution in other edible insects are in needed. Finally, there is concern about the potential dangers of MPs, specifically whether and how MP pollution affects human health.

Supplemental Information

Supplemental Information 1 Sample type, weight wet, MPs and color

Click here for additional data file.

Additional Information and Declarations

Competing Interests

Author Contributions

Data Availability

The authors declare there are no competing interests.

Witwisitpong Maneechan and Taeng On Prommi conceived and designed the experiments, performed the experiments, analyzed the data, prepared figures and/or tables, authored or reviewed drafts of the paper, and approved the final draft.

The following information was supplied regarding data availability:

The raw data are available in the Supplemental File.

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
