# Peer review of "Occurrence of microplastics in edible aquatic insect Pantala sp. (Odonata: Libellulidae) from rice fields"

_PeerJ, doi:10.7717/peerj.12902_

## Round 0.1 · original submission · Major Revisions

Your manuscript was evaluated by three experts in the field. In doing so, some problematic aspects were identified that must be resolved before the manuscript can be accepted. The most crucial issues are the following:

Low sample size: Obviously, only 9 samples (3 times 3 replicates) were analysed with FTIR. This makes a valid statistical analysis problematic, if not impossible.

Lack of procedural blanks/controls: It is well established that during sample preparation and analysis contamination with MP can occur. The degree of such contamination can only be assessed by using procedural blanks/controls which are lacking in your study.

The reviewers underline that more clarity and detail in the method description are needed to show exactly what has been done. Example: How many particles were analysed with FTIR. There are also doubts regarding the correctness of your polymer identification by FTIR.

Additionally to the reviewers' comments please note that the detection of cellulose (natural polymer) and PEGs (liquid polymers) does not confirm the presence of microplastics.

Reviewer 1 ·

Basic reporting

The article structure is good but a paragraph on contamination control should be added. Also, the English should be improved. Literature references are sufficient.

L. 36: do you mean FTIR or µFTIR throughout the manuscript?

L. 39: treatment is not a suitable term in this context

L. 60: there are much more references, please write …e.g. Wagner…

L. 62: The English should be improved throughout the manuscript. For instance, it should be 'contaminates' here.

L. 72: 'contain' instead of 'contains'

In the file 'Raw_data_MPTs_in_edible_aquatic_insects.xlsx' there are some Thai characters/letters that cannot be understood by all readers.

Figure 3: Please, define rods.

Regarding Figure 6: what is the red spectrum and what is the gray spectrum supposed to mean? Is one of them a reference?

Experimental design

The study 'Occurrence of microplastics in edible aquatic insect Pantala sp. (Odonata: Libellulidae) from rice fields' analyses microplastic loads in the bodies of 45 (partly pooled) dragonfly larvae from Thailand. Those larvae are regularly eaten by humans.

Although it is important to gain more insight into microplastic pollution in Thailand, I recommend rejection of the manuscript. The authors are providing extremely little information on microplastics in invertebrates. They recorded 247 organisms from different taxa but only analyzed 45 (pooled) dragonfly larvae (resulting in 3 replicates each) for microplastics. The study seems incomplete and they basically analyzed 9 samples (three times 3 replicates). This is not sufficient for a valid statistical analysis. The authors also did not use any blanks/controls to test for potential microplastic contamination coming from their chemicals and they do not mention how many particles they analyzed with FTIR. Regarding FTIR, I seriously doubt that their polymer identification is correct. The substances (such as hydroxyethyl cellulose) are not reported in microplastic studies. Also, are the 4 mentioned substances (hydroxyethyl cellulose, poly(ethylene glycol) methyl ether, poly (ethylene glycol) tetrahydrofurfuryl ether, and poly (ethylene glycol) tetrahydrofurfuryl ether (methyl vinyl ether)) even plastic?
The spectra (Figure 6) are also not really clear and might be masked by residues stemming from the digested organisms. That should be looked at again.

Please, find more detailed comments below:

L. 65: microplastics can also be quickly egested by organisms

L. 131: Please provide the mean weight ± SE or SD to show that all 45 larvae had similar weights.

L. 135: When pooling the organisms, information on microplastic loads per individual gets lost. It would have been better to analyse the larvae individually. Also, that leaves you with only 3 replicates for statistics which is not enough.

L. 135: Please, add more information on contamination control. For instance, how did you avoid contaminating the dragonfly organs when dissecting the organisms?

L. 142 to 147: Did you check whether your chemical treatment would digest any microplastics? Also, a control treatment with only chemicals and no organisms seems to be missing. Such a control is essential to exclude any microplastic contamination stemming from the lab chemicals.

L. 149 and 150: Potassium formate is used to separate microplastics from sand grains but not from soft tissues.

L. 154: which sediment are you talking about since you did not take any sediment samples?

L. 154: Why did you use a plastic (nylon) membrane for filtration? That could contaminate your samples.

L. 160 and 161: Can this spectroscope be used for analyzing microscopic particles?

L. 161: How many particles were analyzed with FTIR?

L. 165: What about homogeneity of variances as an assumption for an ANOVA?

L. 256: Were there more microplastics in the gastrointestinal tracts than in the whole bodies?

L. 222 to 226 and Figure 6: I honestly doubt that polymers were correctly identified. The measured spectra are not very clear and it rather seems like there are residues (from e.g. the organisms) that disturb the FTIR measurement. Please, check that again. Also, the substances (such as hydroxyethyl cellulose) that you mention here are not really reported in any microplastic study. Also, are the 4 mentioned substances (hydroxyethyl cellulose, poly(ethylene glycol) methyl ether, poly (ethylene glycol) tetrahydrofurfuryl ether, and poly (ethylene glycol) tetrahydrofurfuryl ether (methyl vinyl ether)) even plastic?

Validity of the findings

L. 188: It is not really clear to me why the authors recorded 247 organisms from different taxa but only analyzed 45 (pooled) dragonfly larvae (resulting in 3 replicates each) for microplastics. The study seems incomplete.

L. 260 to 263: that is interesting but irrelevant for the study

I also do not agree with the authors that their study serves as a baseline since there is not a lot of data which is provided by the authors.

·

Basic reporting

No comment

Experimental design

Please see full comments in additional comments below.

Validity of the findings

Please see full comments in additional comments below.

Additional comments

General comments:
The manuscript “Occurrence of microplastics in edible aquatic insect Pantala sp. (Odonata: Libellulidae) from rice fields” provides an assessment of microplastic concentrations in an edible aquatic invertebrate widespread among rice fields in Asia. The manuscript is interesting but needs a few changes, specific comments are provided below. But generally there could be more clarity and detail in the methods to show exactly what has been done by the authors. The writing is very good and only some additional content is necessary. Another large change would be to merge the results and the discussion (or remove the references from the results). All requested changes are provided in more detail below.

Specific comments:
Abstract:
Line 27: Change MPs to MP
Line 29: This was the case a few years ago, but now there is a fair bit of information on freshwater invertebrates, I would rephrase this to say that there is a gap in understanding about pathways of MP contamination into humans (i.e., through edible insects).
Line 31: Remove “for the first time” as it does not add anything to the sentence.
Line 32: State that it is a dragonfly for the general reader’s understanding.
Line 37: Found in all organisms studied?
Line 39: Per individual per treatment? I think treatment is a confusing word here as it is not an experiment. Maybe using preparation method, or something else might be more appropriate.

Introduction:
Line 52: Comma after “environment” and before “is” on the following line.
Line 56: Briefly define primary and secondary MPs.
Line 60: I would say that this is no longer the case with a large amount of research now in freshwaters. See more recent reviews and references therein, such as: Windsor, F. M., Durance, I., Horton, A. A., Thompson, R. C., Tyler, C. R., & Ormerod, S. J. (2019). A catchment‐scale perspective of plastic pollution. Global Change Biology, 25(4), 1207-1221.
Line 64: MPs have already been defined.
Line 66: Spelling of microplastic is incorrect.
Line 72: “Contain” not “contains”.

Materials & Methods:
Line 137: Measured for what? MP concentrations?
Line 141: Were blank samples used to collect and assess airborne contamination?
Line 143: Remove “For”.
Line 158: Provide the model details for the Leica microscope. Also list the different magnifications used.
Line 162: How many scans were used to derive the spectra? What libraries were used for comparison – more detail on this method is necessary.

Results:
I recommend merging the results and discussion. At the moment there are references in the results which is not standard procedure for a paper with a separate results and discussion section. I think the easiest thing to do will be to add the current discussion to the relevant results section and create a joint results and discussion.
Line 175: It was not clear in the methods that different samples were collected during different growing periods. Add more detail to the sampling section of the methods.
Line 186: Try not to use references in the results section as this is interpretation.
Line 207: I presume these are the measurements that I was unsure about in the comment above – make this clear that the measurements for the 15 individuals were physiological/morphological
Line 211: Shouldn’t this be “MPs/replicate” not “MPs/treatment”
Line 216: This is interesting/concerning as you would expect there to be double the amount of MPs in the whole body as it includes both GT and other tissues.
Line 219: Consistent use of either microplastics or MPs throughout the manuscript.

Discussion:
See above comments on merging this with the results.

Reviewer 3 ·

Basic reporting

The authors need to do some major corrections in this regard. Specifically, they need to rephrase many sentences using clear and unambiguous words.

Experimental design

There is no mention of the number of sampling sites from which the specimens were collected.
I see no relevance of the investigated water parameters in this study.

Validity of the findings

Conclusions are not well stated, especially in the abstract section.

Additional comments

Lines 206-217 should be written as a paragraph, and authors should avoid short paragraphs as much as possible.

Annotated reviews are not available for download in order to protect the identity of reviewers who chose to remain anonymous.

---

## Round 0.2 · Minor Revisions

We are almost there. Please consider the aspects raised by the two reviewers, including the remarks of reviewer 3 in the annotated manuscript, in a hopefully final revision of your manuscript.

·

Basic reporting

The authors have made substantial changes to the manuscript based on the comments from 3 reviewers, and the manuscript is now looking much improved.

I do not have any further substantial comments, however, some minor comments are provided in the following subsections.

Experimental design

N/A

Validity of the findings

N/A

Additional comments

Line 267: This reference doesn't pertain to toxicity, it is about trophic transfer and the potential for effects across multiple trophic levels.

Reviewer 3 ·

Basic reporting

No comment

Experimental design

No comment

Validity of the findings

Conclusions are not well stated.

Additional comments

This research is very timely as it would advance knowledge on MP pollution. However, I would like to request that the authors address all the annotated comments in the attached PDF and the additional comments provided below.

Abstract
1. The goal of this study should be rephrased as follows: The goal of this study is to
assess MP contamination in an edible aquatic insect (Pantala sp.) living in rice fields.
This is very important as there are many edible aquatic insects as alluded by the authors.
2. The conclusion of the abstract needs to be fleshed out by mentioning the implications of the ingested MPs for human health.

Introduction
1. Break the second paragraph of the introduction into two and merge them with the preceding and succeeding paragraphs, respectively.
Lines 62-67 should be merged with the first paragraph, while lines 68-84 should form the new second paragraph. This is necessary for a good flow of thoughts.
2. Lines 90-94: Thoughts expressed here make it sound as if the only MP pathway to Pantala sp. is through zooplankton and smaller macroinvertebrates. I suggest that authors rephrase this section since macroinvertebrates are opportunistic feeders and they also interact directly with the water and sediment of their habitat. Microplastics can be picked up directly from water and sediment just as it is possible to ingest such through lower trophic level animals also.
Results
1. I still do not understand the import of physicochemical water quality and the mention/list of other aquatic insects in this study. It is completely out of the scope or aim of the study. I mentioned this point in my former review. I strongly recommend that the authors expunge these aspects from the work and focus on the aim of the study. Should the authors decide to still retain this information, they will need to do a major correction on the introduction and provide a justification for water quality, other insect taxa and finally MP ingestion by Pantala sp.
2. On the abundance of MPs among the three groups: (a) Authors could as well determine MP abundance per wet weight for the whole body, gut only, and body without gut. This could underscore the fact that guts are the reservoir for MPs in many animals (b) If the whole body recorded the highest MP abundance per individual and the body was well-rinsed with ultrapure distilled water so as to get rid of deposited MPs, this should also raise some curiosity and authors can actually suggest reasons why this was so. I know for instance that MPs have been reported in the cases (body enclosures) of caddisflies but I am not certain if they can find their way into their tissues. In the case of Pantala sp, could these MPs have been incorporated in the body integument (cuticle)? I think this is a big research problem that authors should not ignore in this study. It could be a part of the recommendation.

Conclusion
This should be re-written.
(1) Lines 276-277 is not within the scope of this study. The study only quantified MPs in Pantala sp., it did not establish a nexus between MPs and freshwater toxicity.
(2) Line 278-279 should also be rephrased.

Annotated reviews are not available for download in order to protect the identity of reviewers who chose to remain anonymous.

---

## Round 0.3 · accepted · Accept

Thank you for considering the reviewers' comments. However, in the case of the comment of reviewer 3 regarding your abstract (second point) you have done too much of a good thing. I strongly recommend replacing "consume" with "ingest" in the penultimate sentence and deleting the last sentence of the abstract completely. This statement is not supported by data.